# Evidence from a natural experiment that malaria parasitemia is pathogenic in retinopathy-negative cerebral malaria

Dylan S Small[1]*, Terrie E Taylor[2,3], Douglas G Postels[4], Nicholas AV Beare[5,6], Jing Cheng[7], Ian JC MacCormick[5,8], Karl B Seydel[2,3]

[1]Department of Statistics, The Wharton School, University of Pennsylvania, Philadelphia, United States; [2]Department of Osteopathic Medical Specialties, College of Osteopathic Medicine, Michigan State University, East Lansing, United States; [3]Blantyre Malaria Project, Blantyre, Malawi; [4]Department of Neurology and Ophthalmology, College of Osteopathic Medicine, Michigan State University, East Lansing, United States; [5]Department of Eye and Vision Science, University of Liverpool, Liverpool, United Kingdom; [6]St. Paul's Eye Unit, Royal Liverpool University Hospital, Liverpool, United Kingdom; [7]Department of Preventive and Restorative Dental Sciences, University of California, San Francisco, San Francisco, United States; [8]Malawi-Liverpool-Wellcome Trust Clinical Research Programme, Blantyre, Malawi

*For correspondence: dsmall@wharton.upenn.edu

Competing interests: The authors declare that no competing interests exist.

**Abstract** Cerebral malaria (CM) can be classified as retinopathy-positive or retinopathy-negative, based on the presence or absence of characteristic retinal features. While malaria parasites are considered central to the pathogenesis of retinopathy-positive CM, their contribution to retinopathy-negative CM is largely unknown. One theory is that malaria parasites are innocent bystanders in retinopathy-negative CM and the etiology of the coma is entirely non-malarial. Because hospitals in malaria-endemic areas often lack diagnostic facilities to identify non-malarial causes of coma, it has not been possible to evaluate the contribution of malaria infection to retinopathy-negative CM. To overcome this barrier, we studied a natural experiment involving genetically inherited traits, and find evidence that malaria parasitemia does contribute to the pathogenesis of retinopathy-negative CM. A lower bound for the fraction of retinopathy-negative CM that would be prevented if malaria parasitemia were to be eliminated is estimated to be 0.93 (95% confidence interval: 0.68, 1).

## Introduction

Cerebral malaria (CM) is responsible for a substantial proportion of the approximately 500,000 annual malaria deaths and 2,000,000 severe malaria cases (*WHO, 2014*, *WHO, 2015*). CM is defined by the World Health Organization (WHO) as unarousable coma with circulating malaria (*Plasmodium*) parasitemia and no known non-malaria causal explanation (*WHO, 2000*). Based on the presence or absence of malaria-specific retinal changes, CM can be classified as retinopathy-positive (Ret+) or retinopathy-negative (Ret-) (*Lewallen et al., 1999*; *Beare et al., 2006*). Ret- CM is a common and devastating condition – 40% of CM cases in our cohort were Ret- and of these, 12% died and 10% developed neurological problems (*Table 1*). Autopsy data show that children dying of Ret+ CM have a high degree of sequestration of parasitized red blood cells in cerebral vasculature (*Taylor et al., 2004*), considered the pathological hallmark of CM. The pathogenesis of Ret- CM has

**eLife digest** Malaria is a life-threatening disease caused by a parasite that is transferred between people by infected mosquitoes. Most infected individuals suffer flu-like symptoms, but in rare cases malaria can affect the brain, resulting in brain damage, coma or death.

The World Health Organization defines a person as suffering from cerebral malaria if the person is in a coma, has malaria parasites in his or her blood, and has no known alternative cause of the coma. Patients suffering from cerebral malaria are categorized based on whether they have damage to the back of the eyes known as retinopathy. It had previously been found that children who died of "retinopathy-positive" cerebral malaria (i.e. those who had retinopathy) had malaria parasites stuck in small vessels in their brains, which likely caused the coma. By contrast, children who died of "retinopathy-negative" cerebral malaria lacked this parasitic condition, and often also had other infections that can cause a coma, such as meningitis or sepsis.

Because hospitals in many of the areas most affected by malaria often lack the ability to identify what – other than malaria – caused a coma, it was not clear whether malaria parasites influence how retinopathy-negative cerebral malaria develops.

People with certain genetic variants – such as those that underlie sickle cell disease – are protected against the symptoms of malaria infections, and so these variants should also protect against cerebral malaria cases caused by the parasites. Small et al. therefore looked through data that had been collected over several years from people who had been admitted to a hospital in Malawi for cerebral malaria. This revealed that the genetically inherited sickle cell trait is highly protective against retinopathy-negative (as well as retinopathy-positive) cerebral malaria. Therefore, malaria parasites do play a role in a substantial proportion of cases of retinopathy-negative cerebral malaria.

Although Small et al. provide evidence that malaria parasites play a role in retinopathy-negative cerebral malaria, they may not be the only cause of the coma. In the future, the absence of retinopathy could be used as a sign to look for additional factors that contribute to the coma. Currently, all cerebral malaria patients are treated in the same way. Understanding how malaria parasites interact with other illnesses to produce a coma could lead to the development of targeted treatment plans for retinopathy-negative patients.

been a puzzle, in particular the role of malaria parasitemia (*Postels and Birbeck, 2011*). In the only autopsy study among children dying with CM, that we are aware of ( *Taylor et al. (2004)* with follow-up results in *Milner et al. (2015)* and *Barrera et al. (2015)*), among children for whom retinopathy was assessed, 41 of 42 children dying with Ret+ CM had substantial cerebral sequestration of parasitized red blood cells in the cerebral microvasculature (defined as $\geq$23% of cerebral capillaries had sequestration) and mostly lacked other identified potential causes of death besides the malaria parasitemia, whereas all 15 children dying with Ret- CM lacked substantial cerebral sequestration (<23% of cerebral capillaries had sequestration) and mostly had non-malarial etiologies of death (see Appendix 1 for causes of death); these numbers update those in *Taylor et al. (2004)* to include patients enrolled after 2004. Incidental malaria parasitemia is common in people living in areas of high malaria transmission. Therefore, it is possible that at least some children with Ret- CM have a non-malarial etiology of coma and an incidental (asymptomatic) malaria parasitemia. The pathogenesis of Ret-CM is therefore unclear, and the role of malaria parasitemia in the etiology of the acute illness is unknown (*Postels and Birbeck, 2011*). Our aim of the research presented here was to assess the contribution of acute malaria infection in the pathophysiology of Ret- CM.

*Figure 1* depicts three possible pathways to clinically-defined (WHO-defined) CM (*Postels and Birbeck, 2011*). One pathway is to Ret+ CM for which there is evidence that malaria parasites play a primary role. As noted above, at autopsy, Ret+ CM is associated with the sequestration of parasitized red cells in the cerebral microvasculature (*Taylor et al., 2004*). Compared to children with Ret-CM, those who are Ret+ have increased concentrations of *P. falciparum* HRP2, a parasite-produced protein reflecting total body parasite burden (*Seydel et al., 2012*). Ocular funduscopic findings in Ret+ CM mirror the microvascular pathology observed on fluorescein angiography

**Table 1.** Characteristics of study participants at admission, Means ± SD for continuous variables. The proportions of missing data are shown in Appendix 1. There are 3704 community controls, but their characteristics are not shown because only their genotypes and not their clinical characteristics were collected. Bold denotes p-value less than 0.05.

| | Retinopathy Positive CM | Retinopathy Negative CM | Non-Malaria Hospital Controls | p-value, Ret + vs. Ret - | p-value, Ret + vs. Controls | p-value, Ret – vs. Controls |
|---|---|---|---|---|---|---|
| Number of participants | 438 | 288 | 204 | | | |
| Female | 50% | 52% | 43% | 0.54 | 0.11 | 0.05 |
| Age (months) | 40 ± 26 | 44 ± 30 | 46 ± 30 | 0.10 | 0.05 | 0.53 |
| Mid-upper arm circumference (cm) | 14.9 ± 1.6 | 15.0 ± 1.7 | 14.8 ± 1.8 | 0.72 | 0.53 | 0.39 |
| Weight (kg) | 12 ± 4 | 13 ± 5 | 13 ± 6 | 0.37 | 0.29 | 0.74 |
| Height (cm) | 90 ± 16 | 91 ± 17 | 91 ± 20 | 0.30 | 0.40 | 1.00 |
| Temperature (°C) | 38.6 ± 1.2 | 38.4 ± 1.4 | 37.7 ± 1.5 | **0.03** | **<0.001** | **<0.001** |
| Febrile (Temperature≥ 37.5°C) | 81% | 77% | 56% | 0.23 | **<0.001** | **<0.001** |
| Pulse rate – beats/minute | 152 ± 26 | 148 ± 24 | 139 ± 28 | 0.06 | **<0.001** | **<0.001** |
| Respiratory rate – breaths/minute | 47 ± 15 | 45 ± 13 | 45 ± 15 | 0.12 | 0.17 | 0.93 |
| Liver size – cm below costal margin | 2.0 ± 1.9 | 1.5 ± 1.9 | 1.1 ± 1.7 | **<0.001** | **<0.001** | **0.04** |
| Spleen size – cm below costal margin | 1.7 ± 2.1 | 1.6 ± 2.1 | 0.9 ± 1.6 | 0.56 | **<0.001** | **<0.001** |
| Deep breathing | 33% | 25% | 30% | **0.03** | 0.59 | 0.18 |
| Blantyre Coma Score:<br>0<br>1<br>2<br>3<br>4<br>5 | 14%<br>35%<br>49%<br>1%<br>0%<br>0% | 19%<br>38%<br>43%<br>0%<br>0%<br>0% | 28%<br>40%<br>23%<br>3%<br>0%<br>5% | **0.01** | 0.08 | 0.90 |
| CSF opening pressure – mm of water | 176 ± 75 | 152 ± 82 | 176 ± 99 | **0.001** | 0.96 | 0.07 |
| Hematocrit – % | 19.8 ± 6.9 | 28.2 ± 7.5 | 28.1 ± 9.6 | **<0.001** | **<0.001** | 0.86 |
| Platelets | 81,220± 67,219 | 161,600± 124,747 | 248,400± 162,287 | **<0.001** | **<0.001** | 0.86 |
| Malaria parasitemia – parasites/mm$^3$ | 230,500± 321,924 | 180,500± 280,676 | 3,619± 28,917 | **0.03** | **<0.001** | **<0.001** |
| White blood cells | 13,040± 9163 | 13,020± 8923 | 13,930± 9544 | 0.97 | 0.29 | 0.31 |
| Lactate – mmol/liter | 8.6 ± 5.0 | 7.3 ± 4.4 | 5.5 ± 3.9 | 0.05 | **<0.001** | **0.007** |
| Blood glucose – mmol/liter | 6.1 ± 3.9 | 6.8 ± 4.4 | 7.6 ± 5.3 | **0.03** | **<0.001** | 0.05 |
| CSF white cell count – % ≥ 5 | 16% | 20% | 24% | 0.31 | 0.06 | 0.37 |
| Blood culture positive for pathogen | 4% | 2% | 14% | 0.51 | **<0.001** | **<0.001** |
| HIV positive | 18% | 17% | 15% | 0.91 | 0.59 | 0.66 |
| **Outcomes** | | | | | | |
| Discharge outcome:<br>Full recovery<br>Neurological Sequalae<br>Died | 69%<br>10%<br>21% | 78%<br>10%<br>12% | 57%<br>15%<br>28% | **0.003** | **0.01** | **<0.001** |

(**MacCormick et al., 2015**) and are correlated with the severity of sequestration in both the retina and the brain at autopsy (**Barrera et al., 2015**). Two pathways to Ret- CM, pathway (a) and pathway (b) are depicted in **Figure 1**. As noted above, in patients dying with Ret- CM, the cerebral microvasculature does not have substantial sequestered parasitized erythrocytes (<23% of cerebral capillaries have sequestration), plasma concentrations of HRP2 are decreased, and a variety of non-malarial causes of death have been identified (**Taylor et al., 2004**). For Ret- CM, one potential pathway is

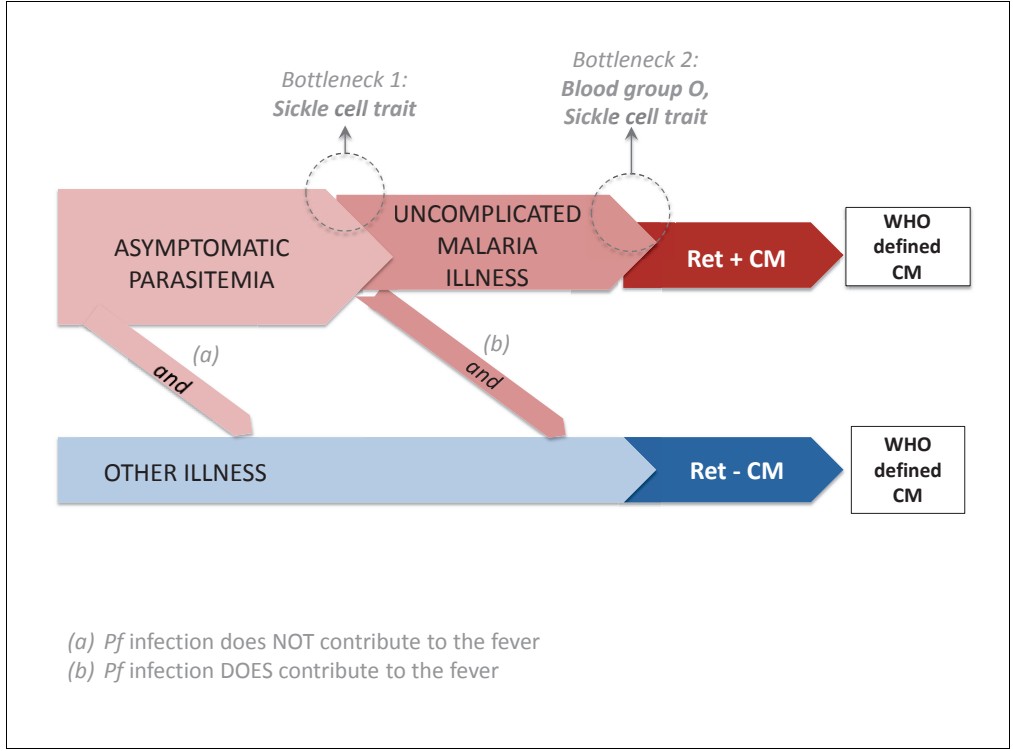

**Figure 1.** Potential pathways to clinically-defined cerebral malaria and genetic bottle necks. There are three potential pathogenetic routes to WHO-defined cerebral malaria (CM). The first, shown in red, is the classical pathway: a malaria infection evolves into retinopathy-positive (Ret+) CM. The second and third possibilities produce retinopathy-negative (Ret-) CM. In (a) the coma is entirely the result of another etiology and the malaria parasitemia is incidental. In (b), the coma is a product of the interaction between the malaria parasitemia and an additional cause (or causes) of coma. Sickle cell trait is underrepresented in patients with Ret+ and Ret- cerebral malaria (CM) because of the bottleneck at the transition between 'malaria infection' (asymptomatic malaria) and 'malaria disease' (uncomplicated malaria). Blood group O is underrepresented in patients with Ret+ CM, but not in those with Ret- CM. Taken together, the results for sickle cell trait and blood group O suggest that some Ret- CM cases occur through pathway (b) (because sickle cell trait is underrepresented in Ret- CM) and that malaria parasites contribute to the pathogenesis of these cases, and that sickle cell trait reduces the pathogenetic potential of malaria infection for Ret- CM but do not provide evidence that blood group O reduces the pathogenetic potential of malaria infection for Ret- CM.

asymptomatic parasitemia and another illness that is sufficient, in and of itself, to produce coma (pathway (a) in *Figure 1*). Another potential pathway is parasitemia leading to uncomplicated malaria illness (e.g., fever) combined with a second insult (innate or acquired), resulting in coma (pathway (b) in *Figure 1*); the two hits (symptomatic malaria+ innate or acquired second factor) result in the clinical syndrome of Ret- CM. A key unanswered question about the pathogenesis of Ret- CM is, are malaria parasites incidental to coma (only pathway (a) exists) or do they play a role in the pathogenesis of Ret- CM (pathway (b) exists) (*Bearden, 2012*; *Postels and Birbeck, 2011*)?

Whether malaria parasites play a role in the pathogenesis of Ret- CM could in principle be tested by a randomized experiment. For example, *Smith (2007)* considered a hypothetical blood-stage malaria vaccine that reduces parasite density by 50%; the vaccine would reduce malaria illness but not the incidence of parasitemia. If such a vaccine existed, then a way to test whether malaria parasites are pathogenic in Ret- CM would be to randomize a large number of children to either (i) the blood-stage vaccine or (ii) placebo. If malaria parasites are never pathogenic in Ret- CM, then we would expect no difference in Ret- CM because the blood stage vaccine would fail to prevent the cause of the development of the Ret- CM whereas if malaria parasites are sometimes pathogenetic in Ret- CM in a way that requires the development of uncomplicated malaria illness, then the blood stage vaccine would prevent the development of Ret- CM in some cases. Such an experiment is not

currently feasible because no blood-stage vaccine has reached a Phase III trial (*Miura, 2016*) and even if an effective blood-stage vaccine was developed, the experiment would require a huge sample size to have power to detect a change in Ret- CM rates.

Though a randomized experiment with a blood-stage malaria vaccine that would have power to detect a difference in Ret- CM rates is not currently feasible, nature provides traits that protect against malaria illness in a random way through genetic inheritance. The general approach of using genetic variation to construct natural experiments is called Mendelian randomization (*Smith and Ebrahim, 2003*). The sickle cell trait (HbAS) – inheritance of one abnormal allele of the betaglobin gene – protects against symptomatic malaria (*Modiano et al., 2001*; *Taylor et al., 2012*; *Williams et al., 2005*; *Willcox et al., 1983*). Thus, a person who inherits one abnormal allele of the betaglobin gene has an antimalarial biochemical protection provided by nature. A second inherited trait which affects susceptibility to malaria illness is blood group O (BGO), which protects against CM compared to groups A, B or AB (*Cserti and Dzik, 2007*; *Malaria Genomic Epidemiology Network, 2014*). Analogous to the randomized trial described above, possession vs. lack of a malaria protective trait assigns children to an arm in which some malaria illness is prevented vs. not prevented. For possession vs. lack of a trait to be fully analogous to the blood stage vaccine randomized trial described above, possession of the trait cannot affect malaria parasitemia incidence just like assignment to the blood-stage vaccine arm in the randomized trial does not affect malaria parasitemia incidence. If the trait affects malaria parasitemia incidence, then it could decrease the Ret- CM rate not because it decreases coma but because the WHO definition of Ret- CM requires malaria parasitemia; the natural experiment induced by the trait would then be biased for assessing the effect of the trait on coma in the same way that a randomized trial is biased if the treatment could affect the measurement of the outcome (WHO-defined Ret- CM) without affecting the true outcome of interest (coma without malarial retinopathy). For both BGO and HbAS, it is plausible that the traits do not protect against malaria parasitemia incidence as systematic reviews have not found consistent evidence for protection (*Uneke, 2007*; *Taylor et al., 2012*); we will assume no protection for our main analysis but do sensitivity analyses that allow for protection. Under the assumptions that a trait does not affect malaria parasitemia incidence and the trait is randomly assigned, then if the trait decreases the probability of developing both Ret+ and Ret- CM, this suggests that malaria parasites contribute to the pathogenesis of both conditions. If the trait decreases the probability of developing Ret+ CM but not Ret- CM, this would suggest that malaria parasites are pathogenetic for Ret+ CM but are either not pathogenetic for Ret-CM or the trait affects an aspect of disease not causal to the development of Ret- CM.

## Results

Using data gathered from 1996 to 2007 in a study of CM pathogenesis in Blantyre, Malawi (*Taylor et al., 2004*; *Seydel et al., 2015*) as well as the MalariaGEN consortium (*Malaria Genomic Epidemiology Network, 2008*), we compared children with CM to two types of controls – (1) community controls; (2) hospital controls, children who were admitted to the Paediatric Research Ward with a known non-malarial cause of illness – meningitis, non-malarial anemia or other non-malaria illness. *Table 1* shows admission characteristics of the hospitalized participants. The Blantyre coma score is statistically significantly higher in Ret+ CM patients than Ret- CM patients, but we do not regard the difference as clinically significant. In general, the patients with Ret+ CM were more severely ill than those with Ret- CM (higher lactate, more deep breathing and a higher chance of death). The malaria illness is more severe in Ret+ CM than Ret- CM patients (lower platelet count and more anemia). The higher opening CSF pressures in Ret+ CM patients compared to Ret- CM patients suggests a higher proportion of Ret+CM patients have increased brain volume. Comparing CM cases to non-malaria controls, as expected, laboratory abnormalities associated with malaria infection (e.g. low hematocrit and platelet count) were more frequent in the CM cases.

For each trait $t$ ($t$=HBAS or blood group O (BGO)), we test the null hypothesis ($H_0^t$) that the trait frequency is the same in controls and true Ret- CM cases vs. the alternative hypothesis ($H_a^t$) that the trait frequency is higher in controls than true Ret- CM cases. Here, true Ret- CM refers to Ret- CM measured without error; in the actual data, retinopathy status may be measured with error and this measurement error is taken into account in the inferences (Materials and methods). Under a model in which the trait does not affect other potential contributors to Ret- CM besides malaria and does

not affect which CM cases are admitted to the Paedeatric Research Ward (as compared to dying before reaching the Ward or recovering before being referred to the Ward), the null hypothesis $H_0^t$ implies that malaria parasitemia is an incidental finding in children with Ret- CM and/or the trait affects an aspect of disease not causal to development of Ret- CM, while the alternative hypothesis $H_a^t$ implies that malaria parasitemia is necessary for some Ret- CM cases and the trait reduces the pathogenetic potential of malaria infection for Ret- CM. Note that if either $H_0^{HbAS}$ or $H_0^{BGO}$ is false, this implies that malaria parasitemia plays a pathogenetic role in at least some Ret- CM cases.

We calculated HbAS and BGO proportions in study subjects and made inferences about odds ratios (*Table 2*) and tested $H_0^{HbAS}$ and $H_0^{BGO}$. For both HbAS and BGO, non-malaria hospitalized controls did not differ from community controls (HbAS p-value=0.86; BGO p-value=0.83); therefore, subsequent analyses combined the control groups. Controls had a higher proportion of HbAS than true Ret- CM patients (odds ratio: 14.33, 95% CI: 3.21, 257.24) and true Ret+ CM patients (odds ratio: 1223.22, 95% CI: 9.87,∞). For BGO, the controls were comparable to true Ret- CM patients (odds ratio: 1.03, 95% CI: 0.83, 1.29) but higher than true Ret+ CM patients (odds ratio: 1.23, 95% CI: 1.01, 1.50). There is strong evidence to reject $H_0^{HbAS}$ (p-value<0.0001) but not $H_0^{BGO}$ (p-value=0.79); these results are insensitive to different plausible assumptions about the false discovery rate and false omission rate for malarial retinopathy (Appendix 1). Taken together, these tests suggest that

**Table 2.** The top panel displays sickle cell trait (HbAS) proportions in retinopathy-positive (Ret+) cerebral malaria (CM), retinopathy-negative (Ret-) CM and control groups. The bottom panel displays ABO blood group gene proportions in Ret+ CM, Ret- CM and control groups. The last two rows of each panel display the odds ratios comparing controls to true Ret+ and true Ret- CM groups, which account for the fact that there is measurement error in observed retinopathy status (false discovery rate = 0.07 and false omission rate = 0.05).

|  | Ret+ CM | Ret- CM | Non-malaria hospital controls | Community controls |
|---|---|---|---|---|
| Sample size | 438 | 287 | 192 | 3657 |
| HbAS* | 0 | 1 | 8 | 175 |
| HbAA | 437 | 286 | 184 | 3482 |
| Proportion of HbAS | 0 | .003 | .042 | .048 |

|  | Odds ratio (95% CI) |
|---|---|
| Non-malaria hospital controls vs. community controls | 0.87 (0.36, 1.78) |
| Controls vs. true Ret- CM | 14.33 (3.21, 257.24) |
| Controls vs. true Ret+ CM | 1223.22 (9.87, ∞) |

|  | Ret+ CM | Ret- CM | Non-malaria hospital controls | Community controls |
|---|---|---|---|---|
| Sample size | 433 | 286 | 199 | 3543 |
| Blood Group O | 175 | 135 | 96 | 1739 |
| Blood Group A, B or AB | 258 | 151 | 103 | 1804 |
| Proportion of Blood Group O | .404 | .472 | .482 | .491 |

|  | Odds ratio (95% CI) |
|---|---|
| Non-malaria hospital controls vs. community controls | 0.97 (0.72, 1.30) |
| Controls vs. true Ret- CM | 1.03 (0.83, 1.29) |
| Controls vs. true Ret+ CM | 1.23 (1.01, 1.50) |

* HbAS (sickle cell trait) means that that the person has one normal and one abnormal copy of the hemoglobin beta gene. HbAA means the person has two normal copies of the hemoglobin beta gene.

malaria parasitemia is pathogenic for a proportion of Ret- CM cases. Sickle cell trait protects against Ret- CM, but blood group O does not.

In Materials and methods, we formulate a sufficient-component cause model (*Rothman, 1976*) based on *Figure 1* and describe how to make inferences about the fraction of Ret- CM cases that are due to pathway (b) in *Figure 1*, i.e., the malaria parasitemia attributable fraction of Ret- CM (the fraction of Ret- CM cases that would be prevented if malaria parasitemia were eliminated [*Benichou et al., 1998*]). The fraction itself cannot be estimated without strong biological assumptions (*Greenland and Robins, 1988*), but a lower bound can be estimated under plausible assumptions. *Table 3* shows inferences for this lower bound under the main model that assumes the traits do not protect against malaria parasitemia incidence and sensitivity analyses. Under the main model, the lower bound is estimated to be .93 with 95% confidence interval (.68, 1). For the sensitivity analyses, although the systematic review of *Taylor et al. (2012)* found no consistent evidence that HbAS reduces malaria parasitemia incidence, some studies reviewed found protection and we consider sensitivity analyses that allow for a small amount of protection (10%) and the largest amount of protection found in all the studies reviewed (41%) (*Ntoumi et al., 1997*). Also, the sensitivity analyses vary the false discovery rate and false omission rate between 0 and the upper bound estimated in Materials and methods. Under all scenarios considered, we found evidence for a substantial contribution of malaria parasites to the pathogenesis of Ret- CM with lower 95% confidence bounds ranging from .37 to .77 and point estimates for the lower bound ranging from .86 to .95.

## Discussion

We have studied a natural experiment that alters the level of malaria illness and found evidence that children with genetic traits associated with resistance to malaria illness are underrepresented in admissions with both Ret+ and Ret- CM (HbAS) or in admissions with Ret+ CM only (BGO).

**Table 3.** Inferences for lower bound on malaria parasitemia attributable fraction of Ret- CM (fraction of Ret- CM cases that would be prevented if malaria parasitemia were to be eliminated) under the sufficient-component cause model based on *Figure 1* presented in Materials and methods. Inferences under the main model and sensitivity analyses that vary the effect of HbAS on malaria parasitemia incidence rate, the false discovery rate (*FDR*) and the false omission rate (*FOR*) for malarial retionopathy.

| Effect of HbAS on malaria parasitemia incidence rate | FDR | FOR | Lower bound on malaria parasitemia attributable fraction of Ret- CM Estimate (95% CI) |
|---|---|---|---|
| Main Model | | | |
| No Effect | .07 | .05 | .93 (.68, 1) |
| Sensitivity Analyses | | | |
| Reduce 10% | .07 | .05 | .92 (.64, 1) |
| Reduce 41% | .07 | .05 | .88 (.46, 1) |
| No Effect | .30 | .11 | .94 (.75, 1) |
| Reduce 10% | .30 | .11 | .94 (.72, 1) |
| Reduce 41% | .30 | .11 | .91 (.58, 1) |
| No Effect | 0 | .11 | .92 (.62, 1) |
| Reduce 10% | 0 | .11 | .91 (.58, 1) |
| Reduce 41% | 0 | .11 | .86 (.37, 1) |
| No Effect | .30 | 0 | .95 (.77, 1) |
| Reduce 10% | .30 | 0 | .94 (.74, 1) |
| Reduce 41% | .30 | 0 | .91 (.61, 1) |
| No Effect | 0 | 0 | .92 (.66, 1) |
| Reduce 10% | 0 | 0 | .92 (.63, 1) |
| Reduce 41% | 0 | 0 | .87 (.44, 1) |

*Figure 1* shows a model of how HbAS and BGO affect Ret- and Ret+ CM. HbAS protects children with malaria parasitemia from developing uncomplicated malaria illness (e.g., fever) and severe malaria illness (*Taylor et al., 2012*). By contrast, current evidence suggests that BGO has no effect on developing uncomplicated malaria illness (*Uneke, 2007*), but does inhibit the cytoadherence of parasitized red blood cells to endothelial cells in the microcirculation, e.g., by affecting rosetting (*Rowe et al., 2007*) and physical properties of the red cell membrane (*Méndez et al., 2012*), thereby preventing severe malaria illness (*Cserti and Dzik, 2007*; *Migot-Nabias et al., 2000*; *Uneke, 2007*). Consistent with this current evidence, in *Figure 1*, HbAS prevents Ret- CM through pathway (b), which involves two components (uncomplicated malaria illness + other illness), by preventing one of the components, uncomplicated malaria illness, whereas BGO does not protect against Ret- CM through pathway (b) because it does not prevent uncomplicated malaria illness (*Figure 1*).

In *Figure 1* pathway (b), Ret- CM results from an interaction between malaria illness and other illness. While the interactions between malaria parasites and other pathogens are incompletely understood and not well investigated, there is evidence for interaction (*Obaro and Greenwood, 2011*; *Mallewa et al., 2013*; *Postels and Birbeck, 2011*). In studies of sepsis and malaria, the effect of microvascular parasite sequestration on the integrity of the gut mucosa is thought to allow bacterial seeding into the blood stream and hence bacteremia (*Scott et al., 2011*). Another example is that following antigenic challenge from a *Plasmodium falciparum* candidate vaccine, children coinfected with schistosomiasis had lower acquired specific immune responses than those not infected (*Diallo et al., 2010*).

Another way in which malaria parasitemia could play a causal role in Ret- CM besides pathway (b) in *Figure 1* is that the parasitemia could be the sole cause of coma. *Taylor et al. (2004)*'s autopsy study results suggest that among children dying from Ret- CM, malaria parasitemia is typically not the sole cause of death (see Causes of Death in Autopsy study in Appendix 1), but it is possible that in survivors from Ret- CM, malaria parasitemia is the major contributor to acute illness. Whether malarial retinopathy is present or absent in CM could be affected by factors such as the child's genome, the parasite's genome and the child's previous exposures to malaria (*Postels and Birbeck, 2011*). The protection provided against Ret- CM by HbAS but not BGO could be explained by an interaction between the factor(s) affecting whether malarial retinopathy is present and BGO, e.g., a parasite genotype which causes malarial retinopathy to be absent could interfere with the mechanism by which BGO provides protection against severe malaria.

Our study has several limitations. We only considered the genetic variants of sickle cell trait and blood group because these were the only statistically significant (p<.05) protective variants in the Malawi sample on which malarial retinopathy was measured, but future work could further test the model in *Figure 1* by looking at additional genetic variants that have been found to affect the risk of malaria in other sites in Africa (*Malaria Genomic Epidemiology Network, 2014*). Our study only addresses CM pathogenesis in children. Adult and pediatric CM have important clinical differences and our results may not be generalizable to adults with Ret- CM. Detection of the presence or absence of malarial retinopathy was determined by several ophthalmologists over the 11 year duration of data collection, the sensitivity of detection of retinal changes may have varied between practitioners. Malarial retinopathy was determined on the basis of ophthalmoscopy alone, but since the time of our study, techniques such as optical coherence tomography that may increase sensitivity have been developed (*Joshi et al., 2017*). We measure malarial retinopathy at the time of admission, but patients are admitted at different points on the disease trajectory. Malarial retinopathy can change over time; it doesn't resolve during the 2–4 days of hospitalization but can become worse, which is a poor prognostic sign. Although there are limitations in our study to the sensitivity with which malarial retinopathy is measured, even if a moderate number of Ret- CM cases should be Ret+ CM cases in *Table 2*, e.g., if the false omission rate is .5 and the false discovery rate is 0, there is still strong evidence that HbAS has a protective effect for Ret- CM (p-value=0.007; odds ratio: 6.81, 95% CI: (1.50, ∞)). Our main analysis assumed that possession of HbAS or BGO does not affect malaria parasitemia incidence; however, sensitivity analyses showed that our results were not sensitive to plausible violations of this assumption. Our analysis assumed that HbAS and BGO do not have selection effects on which CM cases are admitted to the Paediatric Research Ward as opposed to dying before reaching the Paedieatric Research Ward or being cured before needing to be referred to the ward; another interpretation of our study is that instead of studying 'cerebral malaria'

as such, we have studied, 'cerebral malaria mild enough to make it to the ward but severe enough not to recover without being taken to the ward.' Some of the non-malaria hospital controls had malaria parasitemia; since children can sometimes develop severe malaria with relatively low levels of parasitemia, it is possible that malaria contributed to the illness in these control participants. Furthermore, older non-malaria hospital controls may have a selection bias for HbAS since they have survived to an older age (*Ackerman et al., 2005*). Our results are robust to considering only the community controls (*Appendix 1—table 4*) which do not suffer from these potential biases of the non-malaria hospital controls.

In summary, we studied a natural experiment that alters the level of malaria illness experienced by children through genetic variation and found evidence that malaria parasitemia is on the causal pathway to a substantial proportion of Ret- CM cases. Our approach of using genetically inherited traits to study CM pathogenesis could be adapted to illuminate the pathogenesis of other diseases.

## Materials and methods

### Setting and study participants

The study sample includes children with WHO-defined CM from 1997 to 2007 who were admitted to the Paediatric Research Ward at Queen Elizabeth Central Hospital, a tertiary referral center and teaching hospital in Blantyre, Malawi. The WHO definition of CM requires coma, circulating *Plasmodium falciparum* parasites, and no other cause of coma evident either by history or physical examination. Patients were enrolled during the rainy season, the time of annual peak incidence of CM. All patients were treated with intravenous quinine as standard antimalarial therapy. Enrollment in the study required explicit written consent from the parent or guardian. There were 947 enrolled patients. Malarial retinopathy was assessed on 726 of these patients and we subsequently limited our analysis to these 726 patients. To assess retinal changes, direct and indirect ophthalmoscopy were performed by an ophthalmologist well versed in findings typical of malarial retinopathy.

We consider two types of controls in the study. The first type are community controls – 3704 children intended to be representative of the populations to which the cases belonged. Community controls were cord blood samples from Queen Elizabeth Central Hospital, mostly from the newborn nursery. The second type of controls are 194 patients who were admitted during the study period to the Paediatric Research Ward at Queen Elizabeth Central Hospital with a non-malarial cause of illness – meningitis, non-malarial anemia or other non-malaria illness.

### Advantages of considering two control groups

Two sources of bias in case-control studies are hidden bias (unmeasured confounding) and selection bias (*Rosenbaum, 1987*, *Rosenbaum, 2002*). Hidden bias occurs when there are unmeasured variable(s) that are associated with the exposure and the outcome. Selection bias occurs when control subjects are selected based on a variable that is affected by the exposure. Different control groups might be subject to different amounts of hidden bias and selection bias. If two such control groups have similar exposure rates, this provides evidence against hidden bias and selection bias (*Rosenbaum, 1987*). Further, if the case group has a different exposure rate than both control groups and the control groups have similar exposure rates, this provides stronger evidence that the exposure has a causal effect on the outcome, rather than the results being biased by hidden bias or selection bias, as compared to a study with a single control group (*Rosenbaum, 1987*).

Community controls are less likely to suffer from selection bias than the hospital controls. Community controls were randomly selected from the same population as the cases. In contrast, hospital controls could suffer from selection bias if the genetic variant of interest was associated with a non-malaria illness – this type of selection bias is known as Berkson's bias (*Westreich, 2012*; *Berkson, 1946*). Although the hospital controls might suffer from more selection bias than the community controls, the hospital controls have the potential advantage that they would reduce hidden bias if there was population stratification such that the genetic variant was associated with an unmeasured population stratification feature (e.g., housing conditions) that caused both malaria and non-malaria illness.

## Genotypes

Participants were genotyped using the Sequenom iPLEX MassARRAY platform (*Malaria Genomic Epidemiology Network, 2014*). The genotyping included 55 SNPs for which there were previously reported associations with severe malaria. Two of the SNPs were found to be statistically significantly (p<.05) associated with cerebral malaria in Malawi – HBB rs334, which encodes the sickle cell trait, and ABO rs8176719, which encodes the blood type (O vs. A, B or AB); see *Figure 1* in *Malaria Genomic Epidemiology Network (2014)*. These two SNPs were used in our analysis.

## Hypotheses

For a given genetically inherited trait, we consider two competing hypotheses for testing whether malaria parasitemia plays a pathogenic role in Ret- CM. We assume a biological model under which the trait does not affect other illnesses that cause Ret- CM and under which the trait does not have a selection effect on which CM cases are admitted to the Paediatric Research Ward vs. dying before reaching the Paediatric Research Ward or being cured before needing to be referred to the ward.

### Null hypothesis

Malaria parasitemia is an incidental finding in children with true Ret- CM and/or the genetic trait affects an aspect of disease not causal to development of true Ret- CM.

Let $\lambda_{t.r+}$ be the probability that a child with true Ret+ CM has a trait $t$ (e.g., $t$ could be the sickle cell trait or blood type O), $\lambda_{t.r-}$ the probability that a child with true Ret- CM has the trait $t$ and $\lambda_{t.co}$ the probability that a control child has the trait $t$. For the observed data, we take into account that malarial retinopathy may be measured with error. Among children with WHO-defined CM, let $FDR$ be the false discovery rate that a child who is found by the examining ophthalmologist to have Ret+ CM actually has Ret- CM and let $FOR$ be the false omission rate that a child who is found by the examining ophthalmologist to have Ret- CM actually has Ret+ CM.

Suppose malaria parasitemia is an incidental finding in patients with true Ret- CM and/or the trait affects an aspect of malaria infection not experienced by Ret- CM patients. Then the probability of the trait would be the same in controls as in patients with true Ret- CM, $\lambda_{t.r-} = \lambda_{t.co}$. If this is the case, we have the following probabilities for observing the trait in the observable groups:

$$P(\text{Control has trait}) = \lambda_{t.co}$$
$$P(\text{Ret} + \text{CM patient has trait}) = (1 - FDR) * \lambda_{t.r+} + FDR * \lambda_{t.co}$$
$$P(\text{Ret} - \text{CM patient has trait}) = FOR * \lambda_{t.r+} + (1 - FOR) * \lambda_{t.co}$$

### Alternative hypothesis

Malaria parasitemia is pathogenic for true Ret- CM and the trait reduces the pathogenic potential of malaria infection for true Ret- CM.

If malaria parasitemia is pathogenic for true Ret- CM and the trait reduces the pathogenic potential of malaria infection for true Ret- CM, then the trait will reduce the probability of true Ret- CM, $\lambda_{t.r-} < \lambda_{t.co}$. We have the following probabilities for observing the trait in the observable groups:

$$P(\text{Control has trait}) = \lambda_{t.co}$$
$$P(\text{Ret} + \text{CM patient has trait}) = (1 - FDR) * \lambda_{t.r+} + FDR * \lambda_{t.r-}$$
$$P(\text{Ret} - \text{CM patient has trait}) = FOR * \lambda_{t.r+} + (1 - FOR) * \lambda_{t.r-}$$

To estimate $FDR$ and $FOR$, we use data from *Beare et al., 2002* who compared two ophthalmologists' concordance in grading malarial retinopathy. As described in the next section, we obtain point estimates of .07 and .05 and upper bound estimates of .30 and .11 for $FDR$ and $FOR$ respectively; the main analyses use the point estimates and additional analyses use the upper bound estimates. For fixed values of $FDR$ and $FOR$, we estimate the parameters $\lambda_{t.r+}$, $\lambda_{t.r-}$ and $\lambda_{t.co}$ by maximum likelihood.

## Estimating false discovery and false omission rates for malarial retinopathy

We use data on 65 patients at Queen Elizabeth Central Hospital in Blantyre, Malawi who were each examined by two ophthalmologists (*Beare et al., 2002*). Malarial retinopathy is diagnosed if any of

the following six signs are present: retinal hemorrhages (RH), macular whitening (MW), foeval whitening (FW), peripheral whitening (PW), vessel changes (VC) and capillary whitening (CW). RH, VC and CW require little observer judgement if they are seen, but they may not be seen, depending on the degree of pupillary dilation and the presence/absence of spontaneous eye movements; for these consider the specificity to be 1 (*Beare et al., 2002*). Identifying MW, FW and PW requires more experience on the part of the observer; MW and FW are more reproducible, and PW is less so (*Beare et al., 2002*).

We first consider the false discovery rate. To estimate an upper bound on the false discovery rate, we assumed that if RH, VC or CW was detected by either ophthalmologist, there was true malarial retinopathy, but if MW, FW or PW was detected without RH, VC or CW, then it was a false discovery; this is likely an overestimate since in some cases where MW, FW or PW was detected without RH, VC or CW, there is likely true malarial retinopathy that was missed by RH, VC and CW. There were 39 patients diagnosed by ophthalmologist 1 with malarial retinopathy, 33 of whom had RH, VC or CW by at least one ophthalmologist and there were 36 patients diagnosed by ophthalmologist 2 with malarial retinopathy, 33 of whom had RH, VC or CW by at least one ophthalmologist. To find a conservative upper bound on the false discovery rate, we consider ophthalmologist 1 and found the 95% Wilson binomial confidence interval (*Wilson, 1927*) based on 6 out of 39 false discoveries, resulting in an 95% confidence interval of (0.07, 0.30), so resulting in an upper bound of 0.30 for the false discovery rate. To find a point estimate for the false discovery rate, we assume that there was true malarial retinopathy if RH, VC or CW was detected by either ophthalmologist or if FW was detected by both ophthalmologists or if MW was detected by both ophthalmologists since FW and MW were found to be reproducible by *Beare et al., 2002*, and then we averaged the resulting point estimates for the false discovery rate for the two ophthalmologists ((4/39 + 1/36)/2), to obtain an estimate of .07.

We next consider the false omission rate. To estimate an upper bound on the false omission rate, we estimate an upper bound on the false omission rate if we were to only use RH, VC and CW to diagnose malarial retinopathy. The actual false omission rate is likely to be at least as small because we also diagnose malarial retinopathy if there's any of MW, FW or PW and we think that the majority of time when we find MW, FW or PW but not RH, VC and CW, there is true malarial retinopathy, whereas a false omission will be relatively rare. We assume that the false positive rate for RH, VC and CW is 0 and use a model similar to model 1 in *Nedelman (1988)*. Specifically, let $\psi$ denote the prevalence of true malarial retinopathy and $\zeta$ the false negative probability that an ophthalmologist will fail to detect RH, VC or CW in a child with true malarial retinopathy. Assume that we have two independent ophthalmologists. Then, the probability that both ophthalmologists will detect at least one of RH, VC or CW is $\psi(1-\zeta)^2$, the probability that one but not the other ophthalmologist will detect RH, VC or CW is $2\psi\zeta(1-\zeta)$ and the probability that both ophthalmologists will detect none of RH, VC or CW is $\psi\zeta^2 + 1 - \psi$. The false omission rate is $FOR = \frac{\psi\zeta}{1-\psi+\psi\zeta}$. We found a point estimate of 0.05 for the false omission rate and a 95% confidence interval (using the percentile bootstrap) of (0, .11). Thus we take .11 as an estimated upper bound for the false omission rate and .05 as a point estimate for the false omission rate, recognizing that it is likely to be an upwardly biased point estimate.

## Statistical inference

We will test the null hypothesis $H_0^t : \lambda_{t.r-} = \lambda_{t.co}$ (the probability of the malaria resistance trait $t$ is the same in true Ret- CM cases as controls) versus the alternative hypothesis $H_a^t : \lambda_{t.r-} < \lambda_{t.co}$ (the probability of the malaria resistance trait $t$ is lower in true Ret- CM cases than controls). To test these hypotheses, we will estimate the parameters under the null and alternative hypotheses by maximum likelihood and use the generalized likelihood ratio test with 1 degree of freedom (*Rice, 2007*). The maximum likelihood estimation takes into account the assumed false discovery rate and false omission rate for malarial retinopathy detection. We will form confidence intervals for the odds ratios of the trait among controls vs. true Ret- CM patients ($[\{\lambda_{t.co}/(1-\lambda_{t.co})\}/\{\lambda_{t.r-}/(1-\lambda_{t.r-})\}]$) and the odds ratio of the trait among controls vs. true Ret+ CM patients $[\{\lambda_{t.co}/(1-\lambda_{t.co})\}/\{\lambda_{t.r+}/(1-\lambda_{t.r+})\}]$ by inverting the generalized likelihood ratio test. We will use Fisher's exact to test whether the odds ratio of the trait differs among the community controls vs. the non-malaria illness controls and construct a 95% confidence interval for this odds ratio.

## Inference for malaria parasitemia attributable fraction for coma among Ret- CM Cases

The malaria parasitemia attributable fraction for coma among Ret- CM cases is the fraction of coma that would be prevented if malaria parasitemia were to be eliminated among Ret- CM cases. We formulate a model using the sufficient-component cause framework (*Rothman, 1976*) and estimate the malaria parasitemia attributable fraction for Ret- CM using this model. A sufficient cause for a disease is a set of conditions that inevitably produces the disease. We assume that Ret+ CM can be represented by one sufficient cause: malaria parasitemia + factors that lead the malaria parasitemia to develop into uncomplicated malaria illness (e.g., lack of immunity) + factors that lead the uncomplicated malaria illness to further progress to complicated malaria illness with coma and malarial retinopathy (e.g., genetic complexity of the malaria infection that overwhelms a child's ability to control the infection). We assume that Ret- CM can be represented by two sufficient causes: (a) malaria parasitemia + another illness that is sufficient, in and of itself, to produce coma without malarial retinopathy; (b) malaria parasitemia + factors that lead the malaria parasitemia to develop into uncomplicated malaria illness + second insult (innate or acquired) that combined with the uncomplicated malarial illness leads to a coma without malarial retinopathy but which would not in and of itself be sufficient to produce coma. These two sufficient causes (a) and (b) for Ret- CM correspond to pathways (a) and (b) to Ret- CM in *Figure 1*. Let $p$ be the proportion of Ret- CM from sufficient cause (b), which is the malaria parasitemia attributable fraction for coma among Ret- CM cases.

Let $r_{t,p}$ be the factor by which the trait $t$ multiplies the risk of malaria parasitemia (i.e., relative risk of malaria parasitemia for individuals with trait $t$ compared to individuals without trait $t$), $r_{t,u}$ be the factor by which the trait multiplies the risk of factors that lead the malaria parasitemia to develop into uncomplicated malaria illness conditional on having malaria parasitemia, $r_{t,c}$ be the factor by which the trait multiplies the risk of factors that lead uncomplicated malaria illness to further progress to complicated malaria illness with coma and malarial retinopathy, $r_{t,a}$ be the factor by which the trait multiplies the risk of another illness that is sufficient, in and of itself, to produce coma without malarial retinopathy in the presence of malaria parasitemia and $r_{t,s}$ be the factor by which the trait multiplies the risk of a second insult that combined with uncomplicated malaria illness leads to a coma without malaria retinopathy but which would not in and of itself be sufficient to produce coma. We assume that $r_{t,u}, r_{t,c} \leq 1$, i.e., that the trait has no effect or a beneficial effect in preventing uncomplicated and complicated malaria.

Let $\lambda_t$ be the proportion of trait $t$ in the population. For a rare disease, this proportion is approximately the proportion of trait $t$ among the controls, $\lambda_t \approx \lambda_{t.co}$. Ret+ CM and Ret- CM are both rare diseases – the vast majority of malaria infections do not progress to cerebral malaria; subsequently we assume $\lambda_t = \lambda_{t.co}$. We then have

$$\lambda_{t.r^+} = \frac{r_{t,p} r_{t,u} r_{t,c} \lambda_{t.co}}{r_{t,p} r_{t,u} r_{t,c} \lambda_{t.co} + 1 - \lambda_{t.co}}$$

$$\lambda_{t.r^-} = (1-p) \frac{r_{t,p} r_{t,a} \lambda_{t.co}}{r_{t,p} r_{t,a} \lambda_{t.co} + 1 - \lambda_{t.co}} + p \frac{r_{t,p} r_{t,u} r_{t,s} \lambda_{t.co}}{r_{t,p} r_{t,u} r_{t,s} \lambda_{t.co} + 1 - \lambda_{t.co}}$$

For our main model, we make the following assumptions: (i) $r_{t,p} = 1$ – the trait has no effect on malaria parasitemia; (ii) $r_{t,a} = 1$ – the trait has no effect on illnesses that are sufficient in and of themselves to produce coma without malarial retinopathy in the presence of malaria parasitemia and (iii) $r_{t,s} = 1$ – the trait has no effect on an insult that combined with uncomplicated malaria illness leads to a coma without malaria retinopathy but which would not in and of itself be sufficient to produce coma. We relax the assumption that $r_{t,p} = 1$ in sensitivity analyses. Under assumptions (i)-(iii),

$$\lambda_{t.r^+} = \frac{r_{t,u} r_{t,c} \lambda_{t.co}}{r_{t,u} r_{t,c} \lambda_{t.co} + 1 - \lambda_{t.co}}$$

$$\lambda_{t.r^-} = (1-p)\lambda_{t.co} + p \frac{r_{t,u} \lambda_{t.co}}{r_{t,u} \lambda_{t.co} + 1 - \lambda_{t.co}}$$

(1.1)

The parameters $\lambda_{t.co}, \lambda_{t.r^+}, \lambda_{t.r^-}$ can be identified from the data and thus *Equation (1.1)* involve two equations in three unknowns ($p, r_{t,u}, r_{t,c}$). The $p$ is minimized (and hence the malaria parasitemia attributable fraction for coma among Ret- CM cases, is minimized) by letting $r_{t,c} = 1$, specifically the

minimizing $p$ solves $\lambda_{t.r^-} = (1-p)\lambda_{t.co} + p\lambda_{t.r^+}$. Let $p_{lb,t}$ represent this lower bound on $p$ based on trait $t$, which is a function of $(\lambda_{t.co}, \lambda_{t.r^+}, \lambda_{t.r^-})$; let $p_{lb,t'}$ represent this lower bound on $p$ based on another trait $t'$, which is a function of $(\lambda_{t'.co}, \lambda_{t'.r^+}, \lambda_{t'.r^-})$ and let $p_{lb} = min(p_{lb,t}, p_{lb,t'})$ be the lower bound on $p$ based on both traits, which is a function of $(\lambda_{t.co}, \lambda_{t.r^+}, \lambda_{t.r^-}, \lambda_{t'.co}, \lambda_{t'.r^+}, \lambda_{t'.r^-})$. We estimate $p_{lb}$ by maximum likelihood and form a 95% confidence interval by inverting the generalized likelihood ratio test for $p_{lb}$.

We conduct sensitivity analyses that allow for $r_{t,p} < 1$. Maintaining the assumptions (ii) $r_{t,a} = 1$ and (iii) $r_{t,s} = 1$, we have that the lower bound on $p$ solves

$$\lambda_{t.r^-} = (1-p)\frac{r_{t,p}\lambda_{t.co}}{r_{t,p}\lambda_{t.co} + 1 - \lambda_{t.co}} + p\lambda_{t.r^+}$$

## Software
The supplemental file rcode_for_paper.R contains R (*R Development Core Team, 2016*) code for replicating the analyses in our paper.

# Acknowledgements

We thank the MalariaGEN consortium and Gavin Band and Dominic Kwiatkowski for facilitating access to the MalariaGEN consortium data. MalariaGEN is supported by the Wellcome Trust (WT077383/Z/05/Z) and by the Foundation for the National Institutes of Health (566) as part of the Bill and Melinda Gates' Grand Challenges in Global Health Initiative. The Resource Centre for Genomic Epidemiology of Malaria is supported by the Wellcome Trust (090770/Z/09/Z). Support was also provided by the Medical Research Council (G0600718; G0600230; MR/M006212/1). The Wellcome Trust also provides core awards to The Wellcome Trust Centre for Human Genetics (075491/Z/04; 090532/Z/09/Z) and the Wellcome Trust Sanger Institute (077012/Z/05/Z and 098051). We thank Sam Wassmer for helpful insights for *Figure 1*. We thank the reviewers and the senior editor for insightful comments that improved the paper.

# Additional information

## Funding

The authors declare that there was no funding for this work.

## Author contributions

DSS, Conceptualization, Software, Formal analysis, Investigation, Methodology, Writing—original draft; TET, KBS, Conceptualization, Resources, Data curation, Investigation, Project administration, Writing—review and editing; DGP, NAVB, IJCM, Investigation, Writing—review and editing; JC, Methodology, Writing—review and editing

## Author ORCIDs

Dylan S Small, http://orcid.org/0000-0003-4928-2646

## Ethics
Human subjects: We analyzed deidentified data from the Blantyre Malaria project and the Malaria Genomic Epidemiology Network which was previously collected prior to our study; informed consent and consent to publish was previously obtained for these data. The use of human subjects in our research project falls under Exemption 4 for Exempt Human Subjects Research.

# Additional files

## Supplementary files
• Source code 1.

## Major datasets

The following previously published dataset was used:

| Author(s) | Year | Dataset title | Dataset URL | Database, license, and accessibility information |
|---|---|---|---|---|
| Malaria Genomic Epidemiology Network | 2015 | Genome-wide study of resistance to severe malaria | https://www.malariagen. net/data/genome-wide- study-resistance-severe- malaria-eleven-popula- tions | Access to this data is available through application to the MalariaGen Independent Data Access Committee. Please see https:// www.malariagen.net/ data/terms-use/ human-gwas-data for more information |

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

## Appendix 1

### Supplemental results

#### Sample sizes and missing data proportions

*Appendix 1—table 1* shows the sample sizes of the Ret+ CM, Ret- CM, non-malaria hospital controls and community controls. *Appendix 1—table 2* shows the proportion of missing data on the genetic traits for each of these four groups. *Appendix 1—table 3* shows the proportion of missing data on the demographic and clinical variables for the Ret+ CM, Ret- CM and non-malaria hospital control groups.

**Appendix 1—table 1.** Sample sizes.

|  | Retinopathy-positive CM | Retinopathy-negative CM | Non-malaria hospital controls | Community controls |
|---|---|---|---|---|
| Sample Size | 438 | 288 | 204 | 3704 |

**Appendix 1—table 2.** Missing data proportions for genetic traits.

|  | Retinopathy-positive CM | Retinopathy-negative CM | Non-malaria hospital controls | Community controls |
|---|---|---|---|---|
| Sickle Cell Trait | .002 | 0 | .054 | .010 |
| Blood Group | .011 | .003 | .025 | .043 |

**Appendix 1—table 3.** Missing data proportions for demographic and clinical variables.

|  | Retinopathy-positive CM | Retinopathy-negative CM | Non-malaria hospital controls |
|---|---|---|---|
| Female | .018 | .021 | .005 |
| Age (months) | 0 | 0 | .005 |
| Mid-upper arm circumference (cm) | .016 | .014 | .054 |
| Weight (kg) | 0 | 0 | 0 |
| Height (cm) | .009 | .024 | .034 |
| Temperature (°C) | 0 | 0 | 0 |
| Pulse rate – beats/minute | .002 | 0 | .010 |
| Respiratory rate – breaths/minute | 0 | 0 | .005 |
| Liver size – cm below costal margin | .009 | .024 | .010 |
| Spleen size – cm below costal margin | .005 | .014 | .010 |
| Deep breathing | .007 | .021 | 0 |
| Blantyre Coma Score: | 0 | 0 | 0 |
| CSF opening pressure – mm of water | .420 | .330 | .623 |
| Hematocrit – % | .009 | .024 | .034 |
| Platelets | .153 | .160 | .132 |
| Malaria parasitemia – parasites/mm$^3$ | .039 | .042 | .025 |

*Appendix 1—table 3 continued on next page*

Appendix 1—table 3 continued

| | Retinopathy-positive CM | Retinopathy-negative CM | Non-malaria hospital controls |
|---|---|---|---|
| White blood cells | .082 | .097 | .118 |
| Lactate – mmol/liter | .653 | .753 | .564 |
| Blood glucose – mmol/liter | .014 | .003 | 0 |
| CSF white cell count – % $\geq$ 5 | .277 | .170 | .275 |
| Blood culture positive for pathogen | .039 | .024 | .059 |
| HIV positive | .144 | .153 | .353 |
| Discharge outcome | 0 | .007 | 0 |

## Causes of death in autopsy study

For children dying with Ret+ CM, among the children with cerebral sequestration, one had pneumonia and one had pneumonia and meningoencephalitis. The one child with Ret+CM but without cerebral sequestration had a likely cause of death of anemia. Six children with Ret+CM and cerebral sequestration had severe anemia which was thought to be due to malaria parasitemia. Among the 15 children dying with Ret- CM, three had Reye's syndrome and pneumonia, three had pneumonia alone, one had pneumonia with spread to meninges, one had pneumonia with meningoencephalitis, one had hepatic necrosis, one had hepatitis, one had skull fractures, one had subdural/intracerebral hematomas, one had left ventricular failure with pulmonary edema, one had septicemia and one had an unknown cause of death.

## Sensitivity to assumptions about false discovery rate and false omission rates for detecting malarial retinopathy

For testing the null hypothesis vs. the alternative hypothesis for the sickle cell trait, the p-value remains below. 0001 as *FDR* and *FOR* are varied up to the upper bounds of .30 and .11 respectively. For blood group O, the p-value increases to 0.94 when *FDR* and *FOR* are set at their upper bounds of .30 and .11 respectively.

## Results using community controls only

*Appendix 1—table 4* shows the inferences from our model for comparing true Ret+ CM to controls and true Ret- CM to controls (Materials and methods) when we use only the community controls.

**Appendix 1—table 4.** Odds ratio comparing community controls to true Ret+ CM and Ret-CM groups, which account for the fact that there is measurement error in observed retinopathy status (false discovery rate = 0.07 and false omission rate = 0.05). The p-values are two-sided p-values for testing that the odds ratio equals 1.

| | Odds ratio (95% CI) | p-value |
|---|---|---|
| HbAS | | |
| Controls vs. true Ret- CM | 14.43 (3.23, 258.94) | <0.0001 |
| Controls vs. true Ret+ CM | 1234.96 (9.93,∞) | <0.0001 |
| BGO | | |
| Controls vs. true Ret- CM | 1.03 (0.83, 1.29) | 0.79 |
| Controls vs. true Ret+ CM | 1.23 (1.01, 1.51) | 0.04 |

