## [Decision Letter]

Thank you for submitting your article "Evidence from a Natural Experiment that Malaria Parasitemia is Pathogenic in Retinopathy-Negative Cerebral Malaria" for consideration by *eLife*. Your article has been favorably evaluated by Prabhat Jha (Senior Editor) and three reviewers, one of whom, Ben Cooper (Reviewer #1), is a member of our Board of Reviewing Editors. The following individual involved in review of your submission have agreed to reveal their identity: Chandy John (Reviewer #3).

The reviewers have discussed the reviews with one another and the Reviewing Editor has drafted this decision to help you prepare a revised submission.

The paper uses data on the distribution of the sickle cell trait and blood type O in different populations to make inferences about the causal role that malaria parasitemia plays in retinopathy-negative cerebral malaria. The work provides strong evidence that malaria parasitemia does contribute to the pathogenesis of retinopathy-negative CM and provides a lower bound malaria parasitemia attributable fraction in retinopathy-negative cerebral malaria.

Summary:

The reviewers agreed that this is potentially important work, though there was a consensus that it would be far more interesting if it could go beyond the focus on hypothesis testing and expand on the work on estimating the attributable fraction (at present only a lower bound on the lower 95% confidence bound is given). It was felt that this was particularly important given that the null hypothesis (no causal role of malaria in ret-CM) is widely seen as a quite an eccentric position in light of autopsy studies (though, admittedly, it is held by some). In addition to this, there was agreement that the work needed to be better motivated with a more complete discussion of relevant previous work in the Introduction and a more complete consideration of alternative hypotheses that might explain the findings in the Discussion. For example, are there other factors (human or parasite) that interact between uncomplicated malaria and blood group O but not HbAS to alter risk of retneg CM?

Essential revisions:

The reviewers agreed that all the points made in the full reviews below should be addressed. In particular:

i) The Introduction should contain a full explanation of relevant autopsy studies with numbers, logic of hypothesis, and the motivation for the work more clearly explained.

ii) Unless there are compelling reasons why this is not possible (if there are, these should be given), the authors should seek to provide point estimates of the attributable fraction and upper and lower uncertainty bounds. This is likely to require additional assumptions (and may well be facilitated by working within a Bayesian framework), in which case analysis of sensitivity to these assumptions/priors should be provided.

iii) The Discussion should be expanded to contain a fuller discussion of alternative hypotheses explaining the data (see in particular comment 3 from reviewer #3) and limitations of the study (see in particular comments from reviewer #2).

iv) Points 4 and 5 raised by reviewer #1 require at the minimum some clarification.

*Reviewer #1:*

The paper uses data on the distribution of the sickle cell trait and blood type O in different populations to make inferences about the causal role that malaria parasitemia plays in retinopathy-negative cerebral malaria. This is a novel and methodologically interesting application of Mendelian randomization ideas which have become very popular in non-communicable disease epidemiology but which are still little used in the communicable disease community. Despite evidence from autopsy studies of patients with retinopathy-negative cerebral malaria (ret-CM), there is still some debate about the role malaria plays in ret-CM so this work does address a real problem. Overall the paper is well-written and does a fairly good job of explaining quite a difficult subject. I believe the findings do represent a substantial advance over previous work.

1) "An autopsy study found […] children dying with ret- CM mostly lacked cerebral sequestration and mostly had non-malarial etiologies of death (Taylor et al. 2004)"

But I think some of the those dying with ret-CM did have cerebral sequestration which seems to contradict the null. Perhaps as well as emphasising the small sample size, this would be a good time to point out that false positives can occur in retinopathy so the small histopathology studies do not provide conclusive evidence against the null. It also suggests to me that rather more prominence should be given to the estimated lower bound on the malaria parasitemia attributable fraction of ret- CM which currently doesn't get a mention in the abstract. To me, this seems like the most important result.

2) I think it's a good idea to illustrate the ideas here with an idealised unethical RCT, but the example given (Introduction, third paragraph) seems to add to the confusion as it depends on patient outcomes of those with ret-CM, which are not considered in this paper. I think a more relevant (if more expensive) impossible RCT would be to randomize a very large number patients with uncomplicated malaria illness to treatment vs. no-treatment. If the null is true there would be no difference in the incidence of ret-CM in the two arms. This leads on to the natural experiment, as HbAS/BGO are effectively doing similar randomisations.

3) The work rests on the assumption that HbAS has no effect on asymptomatic parasitemia (if it did, different prevalences of HbAS might be expected in those with Ret-CM and the non-malaria controls even if malaria played no role in the CM). While this seems approximately true (and if there is an effect it is likely to be small) this assumption may not be completely rock solid. A meta analysis on protection by HbAS against parasitemia (pubmed id 22445352) concluded "Taken together, HbAS does not consistently protect from *P. falciparum* parasitemia." Some studies have found protection though, but not consistently. This is something the authors might want to consider addressing in a sensitivity analysis.

4) The FP and FN rate as defined in the subsection “Hypotheses” are the conventional definitions (i.e. FP = Pr(test +ve | true -ve), FN=Pr(test -ve| true +ve). Equations in the second to last paragraph of the aforementioned subsection and elsewhere: what is written as FP seems to be Pr(true -ve | test+ve) and similarly for FN. So, I think what is written as FP is actually the FDR (false discovery rate) while what is written as 1-FP is the PPV (prob true +ve | test +ve). Similarly, I think what is given as FN is the FOR (false omission rate) and 1-FN is really the NPV. Fortunately, it looks like the calculations later on for "FP" are actually calculating FDR (which is just what is needed). So, I think the numbers are all OK, but the labels need some attention. Note that checking the calculations to see if what is labelled as FN is actually FOR is more difficult and I haven't done this, but it should be checked.

5) Subsection “Inference for Proportion of Retinopathy-Negative CM due to path (a) in Figure 1:” onwards confused me. If p=0 (so all ret-CM comes from path B – that is, all are causally linked to malaria illness) then the proportion of HbAS in the ret-CM patients will be the same as in the control group. This doesn't make sense. I couldn't make sense of the next equation either. Is there a typo here?

*Reviewer #2:*

In a previous autopsy study, the authors observed that cases of cerebral malaria with retinal changes children had cerebral sequestration of parasitized red blood cells in the cerebral microvasculature and no other identified causes of death besides the malaria parasitemia but in those who were retinopathy negative, few had cerebral sequestration and most had non-malaria etiologies of death. Here, the authors use the fact that HbAS (and blood group O) have been found to protect against symptomatic malaria (or cerebral sequestration) to test whether malaria parasitemia contributes to the deaths in CM ret- by comparing the frequencies of sickle cell in CM ret- vs. controls. Studies using sickle cell and other genetic markers have been used in several studies to test hypotheses, but here this is placed on a more formal footing in the framework of Mendelian randomization.

1) The paper frames the problem as being binary in the Introduction and Methods: e.g. "If a given trait decreases the probability of developing ret+ but not ret- CM, this would suggest that malaria parasites are pathogenic for ret+ CM but not ret- CM or the trait affects an aspect of disease not causal to the development of ret- CM".e.g. "we test the null hypothesis that the trait frequency is the same in controls and true ret- CM cases vs. the alternative hypothesis that the trait frequency is higher in control than true ret- cases."

However, it is likely that a proportion of CM ret- and a very high proportion of CM ret+ cases were caused or co-caused by the parasitemia. In both groups, there could be a non-zero proportion of coma caused by non-malaria, unless having CM ret+ excludes other causes.

So, it seems that using significance to test for an association between CM ret- and controls and their sickle cell status does not capture the question. Estimation would be more useful.

The binary stance is softened later, in the Results section, "have found evidence that malaria parasitemia is on the causal pathway to a substantial proportion of ret- CM cases". However, we do not have an estimate of what the proportion is, and this seems to be mostly a guess. A model which estimates the attributable fraction could be considered.

2) The main results are not inconsistent with the hypothesis of the authors, but neither are they strongly supportive of it.

For HbAS, there was a lower frequency of HbAS in the CM cases compared to the controls, which fits with HbAS being protective against symptomatic malaria. However, there is little evidence that the proportions of HbAS for CM ret+ and CM ret- are similar: there is only 1 HbAS in the two groups, so a non-significant result is inevitable, and the confidence interval stretches from 0.18 to infinity.

For blood group O, the authors stated that they would expect to see a higher proportion of children with O in the CM ret- compared to CM ret+ group if parasites did not cause CM ret- symptoms since blood group O protects against adherence and severe malaria in the brain. The estimate is in the right direction but the CI includes 1 (0.88, 1.62), and so the result in inconclusive. There is a fair amount of overlap in the CIs for controls vs. CM ret- and controls versus CM ret+, and comparing the p-values would not be appropriate due to a non-significant result indicating "no evidence of a difference" rather than "no difference", and being partly reliant on the sample sizes, which differ.

Comparability of the groups

3) There is a little discussion of the biases for the controls, but none for the CM cases. More discussion would be helpful. It may perhaps be that some CM cases do not reach hospital because they die more quickly, or that the older controls have selection bias for HbAS.

Both coma score and retinopathy may differ over the course of the illness, and what is seen is just a snapshot at the time of admission.

Can the differences in the Blantyre coma score and other clinical indicators for ret- and ret+ CM in Table 1 be explained?

The controls are mostly from cord blood samples and some from non-malaria hospital admissions. They are combined in the analysis. While this is not unreasonable, it does make it hard to interpret the results.

4) There would be a need to explain the results of the Taylor et al. 2004 study in the light of the findings.

*Reviewer #3:*

This is an elegant study that uses the genetic traits of HbS trait and blood group O, both well described as protecting against severe malaria, to assess whether malaria contributes to the disease process (i.e., development of coma) in children with cerebral malaria (CM) who are retinopathy positive (ret+) vs. retinopathy negative (ret-).

The authors find that HbS is protective against ret+ and ret- CM, while blood group O is protective only against ret+ CM. They posit that this is because HbS protects in the step where asymptomatic parasitemia develops into uncomplicated malaria, while blood group O only protects only in the step from uncomplicated malaria to severe malaria. In this scenario, ret-CM is due to the presence of uncomplicated malaria plus some other factor that leads to coma. Children with HbS are protected from CM, including ret- CM, because they are protected at the stage where asymptomatic parasitemia becomes uncomplicated malaria. However, children with blood group O are protected only against ret+ CM, because in ret-CM, children with blood group O can still develop uncomplicated malaria, and need only the other factor to tip them into coma. They conclude that malaria likely does play a role in some/many cases of ret- CM.

I find the idea behind this innovative assessment compelling. The paper is well written, and the data appear solid, with much larger numbers than most CM studies are able to obtain. So, overall I think this is a well conducted study and of significant interest to the field. I have only a few comments.

1) Some of the "non-malaria" hospital controls had malaria parasitemia. Since children can sometimes develop severe malaria with relatively low levels of parasitemia, how can the authors be sure that malaria did not contribute to illness in these control participants?

2) The authors pioneered assessment of PfHRP2 levels in children with CM, and these were shown to distinguish nicely between RP and RN CM. Do they have PfHRP2 levels on even a subset of these individuals? It would be nice to further differentiate whether, for example, those with low or undetectable PfHRP2 had lower mortality, or less/no protection from HbS, for example.

3) I think the authors' hypothesis is the most satisfactory one, but it seems possible that the results could be explained by protection afforded by HbS against development of severe malaria from uncomplicated malaria that is not afforded by BGO in children with ret- CM. In other words, there may be interaction with ret- CM and BGO such that in ret- children, whatever mechanism BGO provides that affords protection in going from uncomplicated malaria to CM is interfered with or less effective. I am not sure what that factor could be, but it could potentially relate to things like parasite genotype. Have the authors considered this possibility as an explanation?

4) Are there any clinical or other indicators that suggest to the authors what the other factor(s) may be in ret- CM that lead children with uncomplicated malaria to go on to develop coma?

---

## [Author Response]

*Essential revisions:*

*The reviewers agreed that all the points made in the full reviews below should be addressed. In particular:*

*i) The Introduction should contain a full explanation of relevant autopsy studies with numbers, logic of hypothesis, and the motivation for the work more clearly explained.*

We now provide results in the Introduction and Appendix 1 from the autopsy study described by Taylor et al. (2004) that update the results reported in Taylor et al. (2004) to include patients enrolled after 2004. In the Introduction, we now say:

“Autopsy data show that children dying of Ret+ CM have a high degree of sequestration of parasitized erythrocytes in cerebral vasculature, considered the pathological hallmark of this condition (Taylor et al. 2004). […] In the only autopsy study among children dying with CM we are aware of (Taylor et al. 2004; follow-up results in Milner et al. 2015; Barrera et al. 2015), among children for whom retinopathy was assessed, 41 of 42 children dying with Ret+ CM had substantial cerebral sequestration of parasitized red blood cells in the cerebral microvasculature (defined as ≥23% of cerebral capillaries had sequestration) and mostly lacked other identified potential causes of death besides the malaria parasitemia, whereas all 15 children dying with Ret- CM lacked substantial cerebral sequestration (<23% of cerebral capillaries had sequestration) and mostly had non-malarial etiologies of death (see Appendix 1 for causes of death); these numbers update those in Taylor et al. (2004) to include patients enrolled after 2004.”

Taylor et al. (2004) and follow-up papers is the only autopsy study we are aware of among children dying with CM.

We have sought to explain more clearly the motivation for our work in the Introduction. In particular, first, we have added to the Introduction the following motivation for our work:

“Incidental malaria parasitemia is common in people living in areas of high malaria transmission. […] Our aim in the research presented here was to assess the contribution of acute malaria infection in the pathophysiology of retinopathy negative CM. “

Second, as suggested by reviewer 1, we have considered a different, clearer infeasible randomized experiment, which would address our hypothesis of interest that malaria parasites are involved in the pathogenesis of some Ret- CM cases. We then explain how considering the relationship between inherited genetic traits and cerebral malaria provides a natural analogue of this infeasible randomized experiment. See our response to comment 2 of reviewer 1.

*ii) Unless there are compelling reasons why this is not possible (if there are, these should be given), the authors should seek to provide point estimates of the attributable fraction and upper and lower uncertainty bounds. This is likely to require additional assumptions (and may well be facilitated by working within a Bayesian framework), in which case analysis of sensitivity to these assumptions/priors should be provided.*

We have added Table 3 that provides results for point estimates and uncertainty bounds for the attributable fraction. We have rewritten the section in the supplement “Inference for Proportion of Retinopathy-Negative CM due to path (a) in Figure 1” and now name it as “Inference for Malaria Parasitemia Attributable Fraction for Coma among Ret- CM Cases.” The section now formulates in more detail the causal assumptions underlying our inference about the attributable fraction using the sufficient-component cause framework (Rothman, 1976).

*iii) The Discussion should be expanded to contain a fuller discussion of alternative hypotheses explaining the data (see in particular comment 3 from reviewer #3) and limitations of the study (see in particular comments from reviewer #2).*

We have added fuller discussion and alternative hypotheses explaining the data to the Discussion as follows:

“In Figure 1 pathway (b), Ret- CM results from an interaction between malaria illness and other illness. […] The protection provided against Ret- CM by HbAS but not BGO could be explained by an interaction between the factor(s) affecting whether malarial retinopathy is present and BGO, e.g., a parasite genotype which causes malarial retinopathy to be absent could interfere with the mechanism by which BGO provides protection against complicated malaria.”

*iv) Points 4 and 5 raised by reviewer #1 require at the minimum some clarification.*

See our responses to reviewer #1 in which we have made corrections to account for the good points raised by reviewer #1.

*Reviewer #1:*

*[…] 1) "An autopsy study found […] children dying with ret- CM mostly lacked cerebral sequestration and mostly had non-malarial etiologies of death (Taylor et al. 2004)"*

*But I think some of the those dying with ret-CM did have cerebral sequestration which seems to contradict the null. Perhaps as well as emphasising the small sample size, this would be a good time to point out that false positives can occur in retinopathy so the small histopathology studies do not provide conclusive evidence against the null. It also suggests to me that rather more prominence should be given to the estimated lower bound on the malaria parasitemia attributable fraction of ret- CM which currently doesn't get a mention in the abstract. To me, this seems like the most important result.*

We now provide results from the autopsy study described by Taylor et al. (2004) that update the results reported in Taylor et al. (2004) to include patients enrolled after 2004. We now say, “In the only autopsy study among children dying with CM we are aware of (Taylor et al. 2004; follow-up results in Milner et al. 2015; Barrera et al. 2015), among children for whom retinopathy was assessed, 41 of 42 children dying with Ret+ CM had substantial cerebral sequestration of parasitized red blood cells in the cerebral microvasculature (defined as ≥23% of cerebral capillaries had sequestration) and mostly lacked other identified potential causes of death besides the malaria parasitemia, whereas all 15 children dying with Ret- CM lacked substantial cerebral sequestration (<23% of cerebral capillaries had sequestration) and mostly had non-malarial etiologies of death (see Appendix 1 for causes of death); these numbers update those in Taylor et al. (2004) to include patients enrolled after 2004.”

Thanks for the good suggestion to put more emphasis on the malaria parasitemia attributable fraction of Ret- CM. We have now added a sentence to the Abstract, “A lower bound for the fraction of retinopathy-negative CM that would be eliminated if malaria parasitemia were to be eliminated is estimated to be 0.93 (95% confidence interval: 0.68, 1).”

We have also added Table 3 to present the main model and sensitivity analysis results for the malaria parasitemia attributable fraction of Ret- CM and have expanded the description in the results to:

“In Methods, we formulate a sufficient-component cause model (Rothman 1976a) based on Figure 1 and describe how to make inferences about the fraction of Ret- CM cases that are due to pathway (b) in Figure 1, i.e., the malaria parasitemia attributable fraction of Ret- CM (the fraction of Ret- CM cases that would be prevented if malaria parasitemia were eliminated (Benichou et al. 1998)). […] Under all scenarios considered, we found evidence for a substantial contribution of malaria parasites to the pathogenesis of Ret- CM with lower 95% confidence bounds ranging from.37 to.77 and point estimates for the lower bound ranging from.86 to.95.”

*2) I think it's a good idea to illustrate the ideas here with an idealised unethical RCT, but the example given (Introduction, third paragraph) seems to add to the confusion as it depends on patient outcomes of those with ret-CM, which are not considered in this paper. I think a more relevant (if more expensive) impossible RCT would be to randomize a very large number patients with uncomplicated malaria illness to treatment vs. no-treatment. If the null is true there would be no difference in the incidence of ret-CM in the two arms. This leads on to the natural experiment, as HbAS/BGO are effectively doing similar randomisations.*

Thanks for this valuable suggestion. We agree with your comment that the idealized unethical RCT originally given added confusion as it depended on outcomes of those with Ret- CM which are not considered in the paper. As an alternative, we considered the following impossible RCT based on your suggestion – randomize a large number of children to either (i) be followed up and treated with ACT as soon as signs of uncomplicated malaria illness start to develop or (ii) left untreated when signs of uncomplicated malaria illness start to develop. However, in thinking about this trial, we ran into the following difficulty. Once a child is treated with ACT, the ACT will clear the malaria parasites. Consequently, even if the child has a non-malarial infection that would lead to coma without malarial retinopathy, the child’s coma would be classified as a non-malarial coma rather than retinopathy-negative CM because the child no longer has malaria parasitemia. Thus, even if malaria parasites play no causal role in retinopathy-negative CM, we would expect the treatment arm in this trial to have less retinopathy-negative CM.

To address your valuable comment about the problem with our original idealized, unethical RCT, we formulated a different impossible RCT based on a hypothetical blood stage malaria vaccine as follows:

“Whether malaria parasites play a role in the pathogenesis of Ret- CM could in principle be tested by a randomized experiment. […] If the trait decreases the probability of developing Ret+ CM but not Ret- CM, this would suggest that malaria parasites are pathogenetic for Ret+ CM but are either not pathogenetic for Ret-CM or the trait affects an aspect of disease not causal to the development of Ret- CM.”

*3) The work rests on the assumption that HbAS has no effect on asymptomatic parasitemia (if it did, different prevalences of HbAS might be expected in those with Ret-CM and the non-malaria controls even if malaria played no role in the CM). While this seems approximately true (and if there is an effect it is likely to be small) this assumption may not be completely rock solid. A meta analysis on protection by HbAS against parasitemia (pubmed id 22445352) concluded "Taken together, HbAS does not consistently protect from P. falciparum parasitemia." Some studies have found protection though, but not consistently. This is something the authors might want to consider addressing in a sensitivity analysis.*

Thanks for bringing up this important point. We have now made the assumption that HbAS has no effect on asymptomatic parasitemia clear in the Introduction as follows:

“Analogous to the randomized trial described above, possession vs. lack of a malaria protective trait assigns children to an arm in which some malaria illness is prevented vs. not prevented. […] For both BGO and HbAS, it is plausible that the traits do not protect against malaria parasitemia incidence as systematic reviews have not found consistent evidence for protection (Uneke 2007; S. M. Taylor, Parobek, and Fairhurst 2012); we will assume no protection for our main analysis but do sensitivity analyses that allow for protection.”

We now provide a sensitivity analysis to violations of the assumption that HbAS has no effect on asymptomatic parasitemia in Table 3. We discuss this sensitivity analysis in the Results as follows:

“For the sensitivity analyses, although the systematic review of Taylor et al. (2012) found no consistent evidence that HbAS reduces malaria parasitemia incidence, some studies reviewed found protection and we consider sensitivity analyses that allow for a small amount of protection (10%) and the largest amount of protection found in all the studies reviewed (41%).”

*4) The FP and FN rate as defined in the subsection “Hypotheses” are the conventional definitions (i.e. FP = Pr(test +ve | true -ve), FN=Pr(test -ve| true +ve). Equations in the second to last paragraph of the aforementioned subsection and elsewhere: what is written as FP seems to be Pr(true -ve | test+ve) and similarly for FN. So, I think what is written as FP is actually the FDR (false discovery rate) while what is written as 1-FP is the PPV (prob true +ve | test +ve). Similarly, I think what is given as FN is the FOR (false omission rate) and 1-FN is really the NPV. Fortunately, it looks like the calculations later on for "FP" are actually calculating FDR (which is just what is needed). So, I think the numbers are all OK, but the labels need some attention. Note that checking the calculations to see if what is labelled as FN is actually FOR is more difficult and I haven't done this, but it should be checked.*

Thanks for pointing out these errors on our part. We have now corrected them. We have checked that what was labelled as FN is actually FOR and labeled it correctly. We have rewritten the section “Estimating False Discovery and False Omission Rates for Malarial Retinopathy” (and renamed it) to account for the fact that we need to estimate the false discovery and false omission rates rather than the false positive and false negative rates.

*5) Subsection “Inference for Proportion of Retinopathy-Negative CM due to path (a) in Figure 1:” onwards confused me. If p=0 (so all ret-CM comes from path B – that is, all are causally linked to malaria illness) then the proportion of HbAS in the ret-CM patients will be the same as in the control group. This doesn't make sense. I couldn’t make sense of the next equation either. Is there a typo here?*

Thanks for pointing out these errors. We have rewritten this section in the supplement and now name it as “Inference for Malaria Parasitemia Attributable Fraction for Coma among Ret- CM Cases.” The section now formulates in more detail the causal assumptions underlying our inference about the attributable fraction using the sufficient-component cause framework (Rothman, 1976)

*Reviewer #2:*

*[...] 1) The paper frames the problem as being binary in the Introduction and Methods: e.g. "If a given trait decreases the probability of developing ret+ but not ret- CM, this would suggest that malaria parasites are pathogenic for ret+ CM but not ret- CM or the trait affects an aspect of disease not causal to the development of ret- CM".e.g. "we test the null hypothesis that the trait frequency is the same in controls and true ret- CM cases vs. the alternative hypothesis that the trait frequency is higher in control than true ret- cases."*

*However, it is likely that a proportion of CM ret- and a very high proportion of CM ret+ cases were caused or co-caused by the parasitemia. In both groups, there could be a non-zero proportion of coma caused by non-malaria, unless having CM ret+ excludes other causes.*

*So, it seems that using significance to test for an association between CM ret- and controls and their sickle cell status does not capture the question. Estimation would be more useful.*

*The binary stance is softened later, in the Results section, "have found evidence that malaria parasitemia is on the causal pathway to a substantial proportion of ret- CM cases". However, we do not have an estimate of what the proportion is, and this seems to be mostly a guess. A model which estimates the attributable fraction could be considered.*

Thanks for raising these good points. We have now formulated a model for estimating the attributable fraction in the section “Inference for Malaria Parasitemia Attributable Fraction for Coma among Ret- CM Cases.” The results are presented in Table 3. We have added a description of these results as follows:

“In Methods, we formulate a sufficient-component cause model (Rothman 1976a) based on Figure 1 and describe how to make inferences about the fraction of Ret- CM cases that are due to pathway (b) in Figure 1, i.e., the malaria parasitemia attributable fraction of Ret- CM (the fraction of Ret- CM cases that would be prevented if malaria parasitemia were eliminated (Benichou et al. 1998)). […] Under all scenarios considered, we found evidence for a substantial contribution of malaria parasites to the pathogenesis of Ret- CM with lower 95% confidence bounds ranging from.37 to.77 and point estimates for the lower bound ranging from.86 to.95.”

*2) The main results are not inconsistent with the hypothesis of the authors, but neither are they strongly supportive of it.*

*For HbAS, there was a lower frequency of HbAS in the CM cases compared to the controls, which fits with HbAS being protective against symptomatic malaria. However, there is little evidence that the proportions of HbAS for CM ret+ and CM ret- are similar: there is only 1 HbAS in the two groups, so a non-significant result is inevitable, and the confidence interval stretches from 0.18 to infinity.*

*For blood group O, the authors stated that they would expect to see a higher proportion of children with O in the CM ret- compared to CM ret+ group if parasites did not cause CM ret- symptoms since blood group O protects against adherence and severe malaria in the brain. The estimate is in the right direction but the CI includes 1 (0.88, 1.62), and so the result in inconclusive. There is a fair amount of overlap in the CIs for controls vs. CM ret- and controls versus CM ret+, and comparing the p-values would not be appropriate due to a non-significant result indicating "no evidence of a difference" rather than "no difference", and being partly reliant on the sample sizes, which differ.*

Thanks for making these points. We realized from your comment that we had not been clear that our hypotheses of interest are about whether a trait affects Ret- CM and not about how a trait affects Ret- CM vs. Ret- CM. We have removed the comparisons of Ret- CM vs. Ret+ CM.

*Comparability of the groups*

*3) There is a little discussion of the biases for the controls, but none for the CM cases. More discussion would be helpful. It may perhaps be that some CM cases do not reach hospital because they die more quickly, or that the older controls have selection bias for HbAS.*

Thanks for raising these good points. We have added the following discussion of biases for the controls in the limitations and also conducted an additional analysis in Appendix 1—table 4 which only considers the community controls for which the results are similar to the analysis with all controls:

“Our analysis assumed that HbAS and BGO do not have selection effects on which CM cases are admitted to the Paediatric Research Ward as opposed to dying before reaching the Paedieatric Research Ward or being cured before needing to be referred to the ward; another interpretation of our study is that instead of studying ‘cerebral malaria’ as such, we have studied, ‘cerebral malaria mild enough to make it to the ward but severe enough not to recover without being taken to the ward.’ […] Our results are robust to considering only the community controls (Appendix 1—table 4) which do not suffer from the potential biases of the non-malaria hospital controls.”

*Both coma score and retinopathy may differ over the course of the illness, and what is seen is just a snapshot at the time of admission.*

Thanks for pointing this out. We have added this as a limitation, while noting that the results for HbAS having a protecting effect on Ret- CM are somewhat insensitive:

“We measure malarial retinopathy at the time of admission, but patients are admitted at different points on the disease trajectory. […] Although there are limitations in our study to the sensitivity with which malarial retinopathy is measured, even if a moderate number of Ret- CM cases should be Ret+ CM cases in Table 2, e.g., if the false omission rate is.5 and the false discovery rate is 0, there is still strong evidence that HbAS has a protective effect for Ret- CM (p-value =.007; odds ratio: 6.81, 95% CI: (1.50, ∞)).”

*Can the differences in the Blantyre coma score and other clinical indicators for ret- and ret+ CM in Table 1 be explained?*

We have added discussion of the results in Table 1 as follows:

“The Blantyre coma score is statistically significantly higher in Ret+ CM patients than Ret- CM patients, but we do not regard the difference as clinically significant. […] Comparing CM cases to non-malaria controls, as expected, laboratory abnormalities associated with malaria infection (e.g. low hematocrit and platelet count) were more frequent in the CM cases.”

*The controls are mostly from cord blood samples and some from non-malaria hospital admissions. They are combined in the analysis. While this is not unreasonable, it does make it hard to interpret the results.*

We have added an analysis that uses only the community controls (cord blood samples) in Appendix 1—table 4 and the results are similar to those including all controls.

*4) There would be a need to explain the results of the Taylor et al. 2004 study in the light of the findings.*

Thanks for the suggestion. We have added to the Discussion an explanation of how our results relate to the Taylor et al. 2004 study:

“In Figure 1 pathway (b), Ret- CM results from an interaction between malaria illness and other illness[…] The protection provided against Ret- CM by HbAS but not BGO could be explained by an interaction between the factor(s) affecting whether malarial retinopathy is present and BGO, e.g., a parasite genotype which causes malarial retinopathy to be absent could interfere with the mechanism by which BGO provides protection against complicated malaria.”

*Reviewer #3:*

*[…] 1) Some of the "non-malaria" hospital controls had malaria parasitemia. Since children can sometimes develop severe malaria with relatively low levels of parasitemia, how can the authors be sure that malaria did not contribute to illness in these control participants?*

Thanks for raising this good point. We have added this as a limitation, “Some of the non-malaria hospital controls had malaria parasitemia; since children can sometimes develop severe malaria with relatively low levels of parasitemia, it is possible that malaria contributed to the illness in these control participants.” Furthermore, we have added an analysis that uses only the community controls (cord blood samples) in Appendix 1—table 4 and the results are similar to those including all controls.

*2) The authors pioneered assessment of PfHRP2 levels in children with CM, and these were shown to distinguish nicely between RP and RN CM. Do they have PfHRP2 levels on even a subset of these individuals? It would be nice to further differentiate whether, for example, those with low or undetectable PfHRP2 had lower mortality, or less/no protection from HbS, for example.*

This is a good idea to consider. Unfortunately we currently mostly only have the PfHRP2 data for the Ret+ CM patients and not the Ret- CM patients in the cohort years being considered in this paper (1996-2007, which is the period that intersects with when the control data for the Malaria Genomic Epidemiologic Study data was collected). Because there is only one HbAS case among both the Ret+ and Ret- CM cases, we do not think there will be a large enough sample size to study the correlation between HbAS and PfHRP2 with any accuracy.

Regarding the relationship between PfHRP2 and mortality, Dr. Seydel has a paper currently under review that studies this relationship. He finds that there is not a significant correlation between plasma pfHRP2 and mortality but there is a significant correlation between cerebrospinal fluid pfHRP2 and mortality.

*3) I think the authors' hypothesis is the most satisfactory one, but it seems possible that the results could be explained by protection afforded by HbS against development of severe malaria from uncomplicated malaria that is not afforded by BGO in children with ret- CM. In other words, there may be interaction with ret- CM and BGO such that in ret- children, whatever mechanism BGO provides that affords protection in going from uncomplicated malaria to CM is interfered with or less effective. I am not sure what that factor could be, but it could potentially relate to things like parasite genotype. Have the authors considered this possibility as an explanation?*

Thanks for raising this interesting explanation that we had not thought of. We have added discussion of this explanation to the Discussion:

“Another way in which malaria parasitemia could play a causal role in Ret- CM besides pathway (b) in Figure 1 is that the parasitemia could be the sole cause of coma. […] The protection provided against Ret- CM by HbAS but not BGO could be explained by an interaction between the factor(s) affecting whether malarial retinopathy is present and BGO, e.g., a parasite genotype which causes malarial retinopathy to be absent could interfere with the mechanism by which BGO provides protection against complicated malaria.”

*4) Are there any clinical or other indicators that suggest to the authors what the other factor(s) may be in ret- CM that lead children with uncomplicated malaria to go on to develop coma?*

This is an important question. Dr. Postels has an R21 grant in which he is studying factors associated with Ret- CM. He is still analyzing the data, but hopes to have results to report soon.